# Constitution of Long COVID illness, patienthood and recovery: a critical synthesis of qualitative studies

Mia Harrison  ,[1] Tim Rhodes,[1,2] Kari Lancaster[1,3]

¹Centre for Social Research in Health, University of New South Wales, Sydney, New South Wales, Australia
²London School of Hygiene & Tropical Medicine, London, UK
³Goldsmiths, University of London, London, UK

**Correspondence to**
Dr Mia Harrison;
mia.harrison@unsw.edu.au

## ABSTRACT

**Objectives** To investigate the lived experiences of Long COVID.

**Design** Critical interpretive synthesis of qualitative research.

**Data sources** PubMed and Web of Science databases were searched on 14 September 2023.

**Eligibility criteria** Original peer-reviewed qualitative studies describing the experiences of Long COVID were eligible for inclusion.

**Data extraction and synthesis** We used established qualitative synthesis methods to search, screen and manually code the included studies. Critical interpretation methods were used to analyse the data and develop synthetic constructs.

**Results** 68 articles were identified in the first phase of sampling, with 16 studies and 879 participants included in the final synthesis. The analysis of these studies was organised into three thematic constructions of Long COVID: (1) the illness, (2) the patient and (3) recovery. Long COVID was diversely characterised across study approaches, designs and findings but was underpinned by shared diagnostic logics, which shaped the identification and measurement of symptoms. The boundaries between different constitutions of Long COVID in qualitative accounts of illness experience were often imprecise. Slippages between different definitions of Long COVID had implications for patient experiences in relation to diagnosis, help-seeking and care, and expectations of recovery.

**Conclusions** Long COVID is a site of multiple and diverse qualitative interpretation. Accounts of lived experience emphasise the constitutions of illness, patienthood and recovery as situated and emergent. The ongoing context-based negotiation of Long COVID is a defining qualitative feature of the condition. Approaches to researching, diagnosing and developing health interventions must be as adaptive as the varieties of Long COVID lived experience.

## INTRODUCTION

This paper synthesises qualitative research describing the lived experiences of Long COVID. Long COVID is generally characterised in the literature as an uncertain, emerging and complex illness or health condition consisting of a range of possible symptoms that persist following COVID-19 infection.[1–3] As research and awareness of the long-term health effects of COVID-19 have increased, understandings of Long COVID have broadened to encompass diverse illness effects, care needs and trajectories of illness and recovery.[4 5] This has led to the rapid development of clinical knowledge and practice guidelines (coproduced via clinician and patient collaboration) that account for variations and complex presentations of Long COVID.[6] Yet, the classification of Long COVID and expectations of post-COVID-19 recovery remain a subject of contestation and social and medical critique.[4 5 7]

Systematic reviews of Long COVID studies have investigated the characteristics, frequency and heterogeneity of individual Long COVID symptoms (eg, fatigue, cough, chest pain, memory loss, dyspnoea and depression).[1 8–10] These reviews have also noted temporal information (eg, average symptom duration), with one study systematically mapping the temporal persistence of individual symptoms postinfection at 12 weeks and 6 months.[10] Other reviews have synthesised studies focused on specific symptoms

into categories, such as affected bodily systems (eg, cardio-pulmonary, respiratory and multisystem)[11–13] or affected areas of health (ie, physical, mental and social).[14] One review specifically compared the clinical presentation and symptomology of Long COVID and myalgic encephalomyelitis/chronic fatigue syndrome (ME/CFS).[15]

## STRENGTHS AND LIMITATIONS OF THIS STUDY

⇒ This study employed a critical interpretive synthesis method for a review of literature, which enabled the development of new insights about the constitution of Long COVID, extending the goals of the included studies.

⇒ This study developed synthetic constructs that allowed for the analysis of primary evidence and second-order analyses, which enabled critical reflection on how the methods and approaches of the included studies shaped how Long COVID could be understood.

⇒ Included studies were primarily recruited from Long COVID support groups or were developed as qualitative adjuncts to existing Long COVID cohort studies.

⇒ Included studies were sampled primarily from UK and US contexts.

⇒ The included studies primarily used interview and focus group methods.

Reviews have also investigated Long COVID interventions, management and implications for clinical practice.[2 12 16 17] One review grouped post-COVID symptoms into proposed subtypes (eg, non-severe COVID-19 multiorgan sequelae, pulmonary fibrosis sequelae and ME/CFS) and identified potential interventions for each subtype.[18]

Previous review work has tended to focus on the clinical constitutions of Long COVID, with inadequate attention to how Long COVID is experienced and shaped by social relations and contexts. Reviews focusing on qualitative evidence have made greater contributions in such areas, though these are limited in number and scope. One qualitative review of four studies focused on the specific experiences of living with pulmonary sequelae and their implications for nursing practice.[2] Other reviews of qualitative research on living with Long COVID have synthesised the emotional and psychosocial impacts of Long COVID and patient experiences in accessing healthcare and resources.[3 19 20]

One scoping review took a critical sociological approach, synthesising 93 studies (including qualitative studies) to explore clinical definitions and diagnostic constructions of Long COVID.[5] This review focused on the consequences of broadening definitions of 'Long COVID' and argued that overly generous diagnostic boundaries can produce unintended forms of medical ignorance and uncertainty.[5] This analysis contributes to an emerging body of critical literature investigating how Long COVID has come to be classified across medical and social domains.[4 7 21]

Our paper responds to and extends upon critiques regarding broad and fluctuating classifications of Long COVID and their implications for practice and policy.[4 5 7 21] We develop a critical interpretive synthesis of qualitative research describing patient experiences and clinical interpretations of Long COVID. Our synthesis attends to both qualitative accounts of these experiences within the literature and to how Long COVID becomes understood through the framing, methods and analysis of qualitative empirical studies themselves. This enables us to attend to Long COVID in relation with three separate but interconnected concerns: Long COVID as an illness or diagnosis, Long COVID patients and their care needs and experiences of (Long) COVID recovery.

We approach Long COVID as an illness category that is relationally constituted along with emerging knowledge and practices of scientists, clinicians, patients and other stakeholders.[22] Making sense of Long COVID is thus 'a matter […] of negotiation', in which we eschew any goal of a single and definitive description of Long COVID.[23] Our synthesis instead develops tools for understanding the experiences, stakes and implications of Long COVID in different health and social situations and for patients themselves.

## METHODOLOGY

This paper uses critical interpretive synthesis (CIS) methods to review qualitative studies on patient experiences of Long COVID illness, recovery and care. CIS methods prioritise sampling of rich and diverse data and are preferable in cases where the goal of the review is to generate theoretical insights and contributions beyond the findings of the original studies.[24–26] This requires critical, iterative and reflexive approaches to developing a research question and undertaking sampling and analysis, rather than the systematic strategies used in many other review methodologies.[25 27] We outline the steps of our methods below, which build upon an existing, adapted approach to CIS methods.[28]

### Search strategy

We employed a combination of search strategies to identify literature for potential inclusion. First, we conducted searches of PubMed and Web of Science databases on 14 September 2023, using analogous search queries (online supplemental table 1). As CIS methods do not require a systematically generated sample of literature, additional manual searches and citation chaining methods were periodically conducted (until 15 December 2023) to identify other relevant studies for consideration.[28 29] These manual search strategies did not yield additional articles for inclusion.

### Inclusion

We developed four phases of sampling, adapted from an existing CIS approach.[28] These are presented in a simplified form in figure 1; however, in practice, these phases were not discrete or linear steps but instead unfolded during the review process through reflexive, overlapping and iterative practices of testing and analytical refinement. Eligibility for inclusion was kept intentionally broad in Phase I and narrowed as we refined our analytical focus, with additional criteria developed to support a higher overall interpretive value across the final sample.[24] Quality was appraised with the goal of maximising conceptual inclusion[24] using previously developed appraisal prompts[28] (online supplemental table 2). Final inclusion was determined according to principles of conceptual saturation,[24 27] with studies included on the basis of their contribution to theory development. Studies that did not produce any new insights from other included studies were excluded in Phase IV.

Our final sample included 16 studies and 879 participants with Long COVID or similar experiences (not accounting for potential duplication between studies), with most participants living in the UK or USA (table 1). Primary study data were generated from these studies between July 2020 and May 2022. Most studies generated

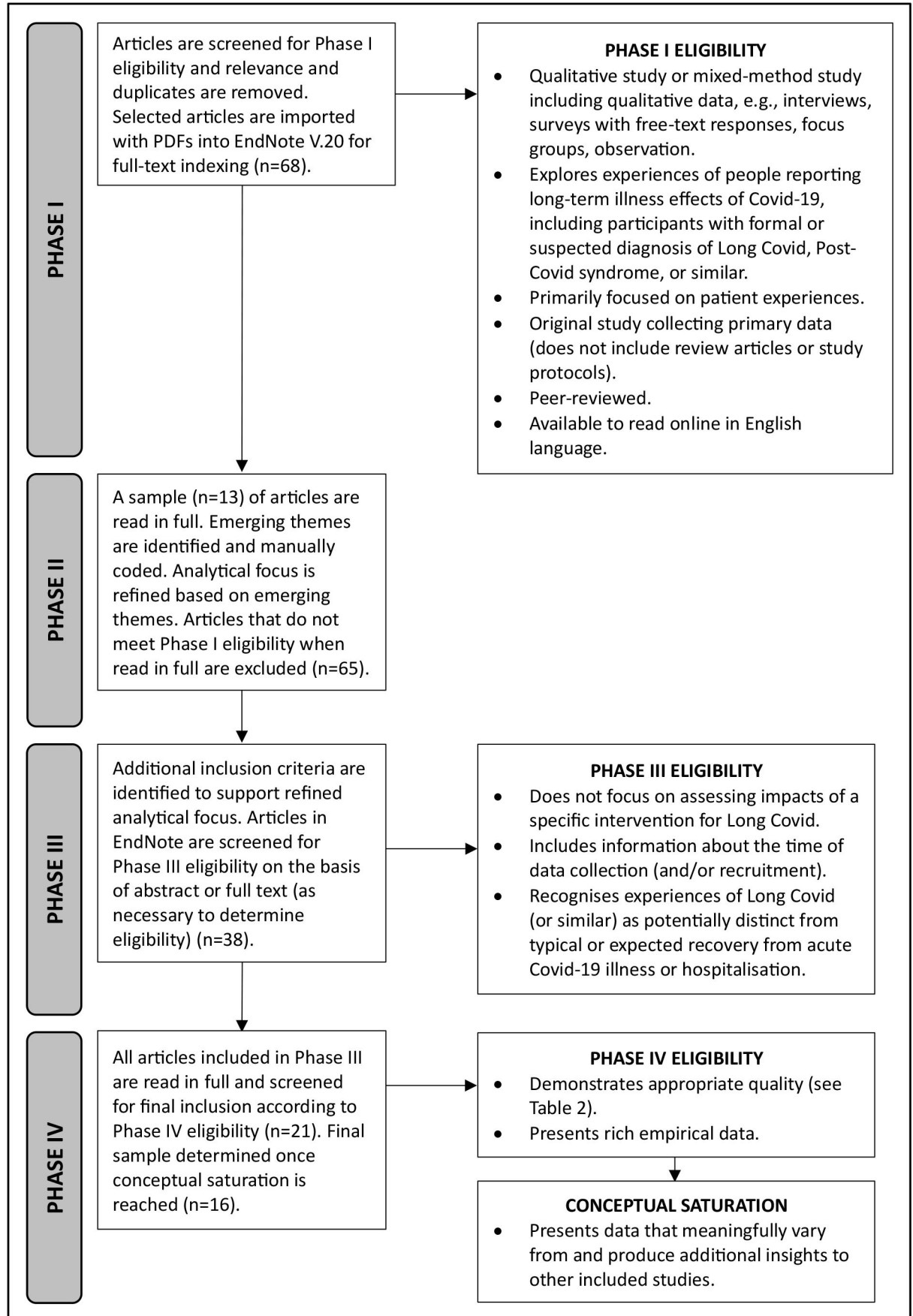

**Figure 1** Flow diagram of phases of sampling for qualitative interpretative synthesis, adapted from existing critical interpretive synthesis approach.[28]

**Table 1** Included studies

| First author (year) | Country | Sample description (N) | Definition of Long COVID to determine study eligibility | Data collection period | Data description |
|---|---|---|---|---|---|
| Aghaei (2022)[30] | USA | Female long haulers (15) | Infected with COVID-19 and having experienced at least one COVID-19 symptom lasting 4 weeks or longer after a COVID-19 diagnosis | April to mid-June 2021 | Interviews |
| Bergmans (2023)[31] | USA | Black patients with Long COVID (15) | Physical or mental health symptoms that lingered over 1 month after an acute COVID-19 infection | May–September 2021* | Interviews |
| Burton (2022)[32] | UK | People who self-reported Long COVID (21) | A positive swab test or antibody test or one or more commonly reported COVID-19 symptoms at illness onset (persistent cough, loss or change in taste or smell or high temperature) and experiencing one or more broader symptoms ≥3 weeks following the onset of their first symptoms | November 2020–September 2021 | Interviews |
| Callan (2022)[33] | UK | People with self-defined Long COVID and people from an online support group dedicated to Long COVID's neurocognitive effects (50) | Not specified | October–November 2020 | Focus groups |
| Chasco (2022)[34] | USA | Patients of a midwestern academic hospital's post-COVID-19 clinic (15) | Persistent health concerns more than 3 months after SARS-CoV-2 infection | July–October 2021 | Interviews |
| Cooper (2023)[35] | UK | People from a West London Long COVID clinical assessment centre (13) | Documented history of SARS-CoV-2 infection and reported persistent or new symptoms (>12 weeks) following COVID-19 | May–July 2021 | Interviews |
| Fang (2023)[36] | UK | People self-identified as experiencing Long COVID symptoms (80) | Reported COVID-19-related symptoms for over 4 weeks, particularly focused on those who indicated having persistent symptoms for over 8 weeks | November 2021–March 2022 | Interviews |
| Maclean (2023)[37] | UK, USA, Netherlands, Canada and Australia | People with (Long) COVID (72) | Not specified | November 2020–March 2022 | Interviews |
| Moretti (2022)[38] | Italy | Women with Long COVID (17) | Experiencing Long COVID symptoms for at least 3 months following confirmed COVID-19 infection; having perceived a significant impact of Long COVID symptoms on quality of life and having consulted multiple healthcare professionals in order to receive a diagnosis | November–December 2021 | Interviews |
| O'Brien (2023)[39] | Canada, USA, Ireland and UK | Adults who self-identified as living with Long COVID (40) | Signs and symptoms that develop during or following an infection consistent with COVID-19 that continue for 12 weeks or more and are not explained by an alternative diagnosis | December 2021–May 2022 | Interviews and participant visual illustrations |
| O'Hare (2022)[45] | USA | Veterans with Long COVID (200) | Documentation of a positive result on a PCR test for SARS-CoV-2 and an International Classification of Diseases-10th revision diagnostic code for Long COVID | February–May 2022 | Electronic health records |
| Ruthforth (2021)[40] | UK | People with Long COVID (114) | Not specified | May–October 2020 | Interviews and focus groups |

Continued

**Table 1** Continued

| First author (year) | Country | Sample description (N) | Definition of Long COVID to determine study eligibility | Data collection period | Data description |
|---|---|---|---|---|---|
| Russell (2022)[41] | USA | Self-identified long-haulers from online COVID-19 communities (20) | Long-term symptoms and effects from COVID-19 with one or more consultations with a healthcare professional about COVID-19 symptoms | March–April 2021 | Interviews |
| Schmachtenberg (2023)[42] | Germany | Long COVID patients (25) | Symptoms that develop during or after an infection consistent with COVID-19, continue for more than 4 weeks and are not explained by an alternative diagnosis | January–May 2022 | Interviews and graphical artist interpretations |
| Taylor (2021)[43] | UK | Doctors with Long COVID (13) | Not specified | July–August 2020 | Interviews |
| Wurz (2022)[44] | International (92% from UK, USA or Canada) | Adults living with Long COVID (169) | Currently experiencing long-term symptoms due to COVID-19 (at least 4 weeks since the acute illness or positive COVID-19 test, with symptoms not predating the acute illness) and having tested positive for COVID-19 or with probable infection (based on an illness mimicking the acute phase of COVID-19, having close contact with a confirmed case or being linked with an outbreak), in line with the clinical case definition for post-COVID-19 condition | February–April 2021 | Open-ended responses in online survey |

*This is the period of participant recruitment, as the period of data collection was not specified.

qualitative data through semistructured interviews or focus groups (of samples ranging from 13 to 114 participants recruited via a combination of surveys, cohorts, social media and clinical or community networks)[30–43] with two studies additionally generating visual data through participant illustration[39] and artist interpretation[42] methods. Other studies generated qualitative data via electronic health records (n=200)[39] and open-ended responses from a larger observational online survey (n=169).[44] Studies used interpretive or descriptive thematic analysis methods, with half of the included studies developing sociologically informed theoretical frameworks.[31 33 36 37 39–42]

### Data extraction and analysis

MH identified and manually coded emerging themes in Phase II of sampling (figure 1) and developed a conceptual argument in collaboration with TR and KL.[22] All three authors have expertise in critical social science approaches to the analysis of qualitative data in health, which informed our theoretical approach to theme generation. These early themes guided the development of additional eligibility criteria, which were used to refine the final sample based on contribution to the conceptual goal of the synthesis.

MH led the development of synthetic constructs to organise the conceptual argument. These constructs were refined in collaboration between the authors through our reflexive and iterative sampling strategies (figure 1). Synthetic constructs allow for both underlying evidence and second-order analyses in included studies to be assembled, critically analysed and transformed, with the goal of developing new conceptual arguments and

analytical insights.[24] We thus approached data analysis of the included studies at two orders: analysis of the primary data presented in the articles and analysis of the interpretations of the primary data developed by the authors of the studies.[24 28]

Following Dixon-Woods *et al*,[24] our analyses of included studies develop a synthesising argument that is grounded in the data yet extends upon the findings of included studies. This enables us to develop new insights and make sense of the research in a novel and explanatory way. Our synthesising argument assembles both our synthetic constructs and second-order interpretations of the data presented by the included studies. This is an established approach to CIS methods,[24 28] but it is especially valuable for our analysis of Long COVID. A key concern has been how Long COVID is studied and understood by academics, clinicians, patients and activists.[4 5 7 21] Our critical interpretation of the study authors' interpretations of Long COVID data is therefore not only an appropriate approach to CIS analysis but also constitutes a crucial line of inquiry for investigating how Long COVID has become known, diagnosed, experienced, researched and cared for as an emerging, uncertain and contested health condition.

### Patient and public involvement

There was no patient or public involvement in this study.

### FINDINGS

Our synthesis investigates the lived experiences of Long COVID. We organise our analysis into three thematic constructions of Long COVID: the illness, the patient

and recovery. Illustrative participant extracts from the included studies, organised by theme and subtheme, are presented in table 2.

## The illness

The first question that studies on Long COVID have addressed is: 'What is Long COVID'? All but two of the included studies were published in 2022 or 2023, yet the complexity of this question remains a concern. Studies defined Long COVID using clinical guidelines and criteria (such as those of the WHO and the UK National Institute for Health and Care Excellence),[32 39 42 43] as well as on the basis of self-identification, noting the role of patient advocacy in the naming of Long COVID.[33 35–37 40 41 44 45] The term Long COVID was generally used, though studies also acknowledged other labels such as 'post-COVID syndrome', 'Post-Acute Sequelae of COVID-19' and 'long haulers'.[30 32–35 38–45] One study explicitly distinguished terms, using 'long COVID-19' to describe 'the lived patient experience' and 'post-COVID-19 syndrome' to describe 'the medically diagnosed condition'.[33]

These working definitions of Long COVID informed participant inclusion criteria, which varied across the studies (table 1). Most studies identified a minimum number of weeks that participants must have experienced persistent or new symptoms after an acute illness (usually 4 or 12 weeks).[30–32 34–36 38 39 42 44] Some also stated preference for confirmed SARS-CoV-2 infection, such as through a positive test or a diagnosis from a clinician.[32 35 44] One study required a formal Long COVID diagnosis from a clinician.[45] A handful of studies determined eligibility through self-identification or did not define Long COVID in their reported eligibility criteria.[33 37 40 43]

All studies described Long COVID as a complex condition consisting of many and varied symptoms. While none of the included studies identified specific symptoms or illness effects as requisites for (or not included within) Long COVID diagnoses, the symptomology and aetiology of Long COVID were primary concerns. All studies we reviewed described specific symptoms experienced by participants, with some studies additionally tracking how common particular symptoms were across the study sample[31 32] or how long symptoms had persisted.[31 32 34 37 39 41 44] Symptoms tended to be described in generally recognisable terms (such as 'brain fog', 'fatigue', 'headaches' and 'shortness of breath'), with a focus on how symptoms were managed and impacted on participants' daily lives.

Yet, in several studies, participants also emphasised the 'strangeness' of their symptoms, suggesting that commonly used terms inadequately described their embodied experiences.[37 40 41 43 44] Recognising these 'strange' symptoms in others allowed people with various illness experiences to collectively identify with the terminology of Long COVID. When these diverse presentations were not widely recognised or accepted (especially early in the pandemic), some participants avoided disclosing symptoms they considered to be 'implausible'.[37 43] Other participants made sense of the complex Long COVID effects through comparisons to health conditions with better understood or more widely accepted biological mechanisms, such as stroke or traumatic brain injury.[33 35 41 43]

Several studies have emphasised that symptoms should not have predated COVID-19 infection or be explained by 'alternative diagnoses'.[39 42 44 45] Yet, in practice, the presentation of Long COVID symptoms was often fuzzy and revealed slippages between descriptions of the symptoms of Long COVID and its extended effects. This was particularly common with psychological and mental health symptoms, with some studies describing symptoms such as anxiety and depression as direct manifestations of Long COVID and others framing mental health impacts as part of a more complex chain of illness effects, for instance, as a consequence of fatigue.[30–32 34 35 38 44] These ambiguous causation chains also intersected with the broader pandemic context.[35] In cases where participants had a pre-existing disability or health condition, experiences of Long COVID were sometimes described as an 'exacerbation' of these conditions, further complicating the actualisation of Long COVID as a distinct and traceable illness.[31 35 39 45]

## The patient

Because of the complexity and diversity of Long COVID symptoms, another challenge in characterising Long Covid was determining who counts as a Long COVID 'patient' (and in what situations). A priori study eligibility criteria inevitably stabilised the constitution of the Long COVID patient to some extent; yet, the Long COVID patient was still heterogeneously defined in studies. For example, some participants referenced an imagined threshold of severity at which a person becomes 'sick enough' to be considered a 'Long COVID patient' (as defined by themselves or by healthcare professionals).[35 41] Overall, however, the 'patient' of Long COVID was presented with two defining features: symptoms that brought about *disruption* in daily life; and a need for *care* as a consequence of symptoms.

Studies generally evidenced 'disruption' in daily life in relation to the capacity of participants to maintain an active social life,[30 32 36 38 39 42 44] return to work,[30 32–34 36 38 41–44] engage in physical activity and exercise[30–32 34–36 42 44] or carry out domestic tasks.[30 32 34 36 37 39 42 44] For example, 'fatigue' was described in several studies in terms of participants having insufficient stamina to play the sports they enjoyed prior to COVID.[31 32 36 42 44] The boundaries of these illness effects were fuzzy and complex, with Long COVID's disruptiveness intersecting with multiple other social and environmental factors; for example, experiences of patienthood might impede the assumed caring responsibilities of motherhood.[30 31]

Many studies framed experiences of seeking care for Long COVID in relation with other experiences of 'chronic illness', 'hidden disability' and 'contested illness', including through comparisons to ME/CFS, HIV-related

**Table 2** Illustrative participant extracts from studies

| Theme | Subtheme | Example quote |
|---|---|---|
| The illness | Complex symptom presentation | 'I had an odd rash for quite a while it kept coming and going… very itchy cough…very mild asthma (….) I just kept putting it down to grief until a couple of months in a friend said, "Look do you think this could be COVID?'[40]<br><br>'The biggest challenge is not knowing when (the episodes are) going to happen, how long they're going to last or how much I can get away with before causing it, if I am truly the cause of one of these episodes'.[39] |
| | Symptoms as inherently strange | 'It's been rather mysterious (…) and it doesn't feel like anything else, it's bizarre'.[43]<br><br>'The fatigue is not exactly tiredness as in sleepiness. It is fatigue as in how a healthy person would feel the day after they ran a marathon'.[44] |
| | Hiding implausible symptoms | 'I'd often get a heart rate of 130, just randomly watching TV. And avoided telling my GP about that. (…) I didn't want to admit it because I think I was worried about it being dismissed as anxiety'.[43]<br><br>'I thought had gotten better, so you know, the phantom smells, the tinnitus, I'd had some strange nerve pain, like a poker in my ear, really bizarre, but it was very, very painful. Strange kind of scalp sensations, as well, but all, all such vague and strange things, I didn't even mention them to, to my GP, because I thought she might think I was completely mad'.[37] |
| | Fuzzy symptom categories and ambiguous causation chains | 'I have constant headaches, that is true, but I don't know if that is because of COVID or just because I'm stressed, it is psychological'.[35]<br><br>'She also relates a history of asthma and chronic cough. Difficult at this time to separate out post-COVID recovery from potential underlying asthma'.[45] |
| The patient | Thresholds of symptom severity for patienthood | 'I feel like it probably doesn't apply to me, because I feel like my symptoms, or whatever I have left, is not as severe as what other people are going through'.[35]<br><br>'I don't want to put any pressure on (the) health service. (It) is quite awful to come and approach a GP and say, "I'm really tired"'.[37] |
| | Disrupted daily life as evidence of patienthood | 'I used to go out dancing twice a week (…) and you think, well, I couldn't even get up, let alone go dancing twice a week… You reach a point where you don't actually remember being well'.[36]<br><br>'I had COVID and 5 months later, I'm still not better.(…) I can't look after my children for long periods of time, or in the initial stages at all, and that created huge problems'.[43] |
| | Health professionals as authenticators of patienthood | 'I have come across a few doctors where I've felt they've made me feel a bit stupid when I'm worrying about certain things… So yes, there's that worrying that people just think you're making it up'.[32]<br><br>'(The doctor) listened to absolutely everything I had to say, every weird symptom that I thought was unrelated. You know, he listened to everything and then would tell me, "Nope, that's all part of it".'[37] |
| | Managing care through self-experimentation | 'I have gone all out on the quackery, which I never thought I would because I've never been that person. I've had acupuncture a few times. I am under a dietician and I'm taking all sorts of weird and wonderful supplements'.[43]<br><br>'I even let my daughter and I let my friend, my mom, wear my Apple Watch for a day. Like, "Let me see if (the high heart rate notification) goes off for you". If it's not going off for you, then why is it going off for me?'[31] |

Continued

**Table 2** Continued

| Theme | Subtheme | Example quote |
|-------|----------|---------------|
| Recovery | Illness and recovery as temporary | 'You don't feel good, but you have to get to work, and then you have kids and family. (…) You are expected to do (it) all'.[30]<br>'I also had two colleagues who complained about me to the staff council, along the lines of "I always have to cover for her because she is sick."'[42] |
| | Non-linear recovery trajectories | 'I liken it to a rollercoaster ride and you're like up and down… there are good days and bad days…. my symptoms are still there but it kind of goes down'.[39]<br>'You feel like shit for a few days (with COVID-19), you start to feel a little bit better, then you feel like shit again (…) it's about a whole new experience of illness. (…) You come to doubt your knowledge of illness and recovery'.[37] |
| | Illness/wellness binaries | 'In March, we thought COVID was something that would either kill me in 3 weeks or I'd get over it… It was May when I started having long-term symptoms. (…) Am I the only one going through this or are there others?'[41]<br>'It's like when somebody has a stroke, and they get partially paralysed.(…) They might regain partial control, like learn to walk again, but their arm might not ever be working again'.[41] |
| | Indeterminacy of acute/Long COVID recovery | 'The Long COVID name to me, feels like you've still got COVID; its like a long version of it. (…) I always say post-COVID, personally, just because usually when talking about things like post-op, you would say, not long-op'.[35]<br>'It's a bit wishy-washy, to me anyway, but also, I think everyone is feeling that they've got slightly different as well. (…) It's a bit of a broad term, isn't it?'[35] |
| | Restorative versus chronic models of illness | 'I would rather (the term Long COVID) than say, chronic COVID, that would be a lot more worrying'.[35]<br>'I consider myself disabled 24/7.(…) I'm always at this minimal state (…) it's not like I go from "oh today I can do all these things". No, there's no day like that'.[39] |

GP, general practitioner.

illnesses or fibromyalgia.[33 34 36–41 43–45] Management of Long COVID via formal health services was described as 'fragmented', insufficiently resourced, 'poorly coordinated' and lacking relevant knowledge or treatment options.[32 37 38 40 41 43 45] Participants expressed frustration at being expected to develop their own treatment plans and at insufficient or ambiguous diagnostic and monitoring technologies for 'making sense' of their Long COVID illness experience.[37–40 43] Participants also reported difficulty in understanding processes for securing referrals to appropriate specialists,[32 37 40] especially if neurological issues made it harder for them to advocate for themselves and explain their experiences.[33] Other barriers to accessing services included requiring formal diagnoses or positive COVID-19 tests[32 37 41] and changing eligibility criteria.[32] Some participants also described feelings of 'guilt' for accessing an under-resourced health system, especially when people were dying from COVID-19.[37 43]

Another care-related concern described in studies was in relation to the diagnostic category of Long COVID. Studies described experiences of medical 'gaslighting' (especially earlier in the pandemic), where participants had to try to 'prove' the reality of their illness and felt that their experiences were often 'minimised' or 'dismissed as anxiety'.[31–33 36–38 40 41 43] Participants emphasised the importance of having 'weird' symptoms 'validated' by a medical professional as part of Long COVID and described this validation as a form of care in itself.[33 35 37 40 43] Participants also reported mixed experiences of care and support from family and friends, noting similar challenges to those of formal care seeking, such as insufficient knowledge and resources or 'disbelief'.[30 31 34 36 38 40] For example, some participants found initial support from friends and family became strained over time,[32] especially when symptoms were 'invisible' or 'unpredictable'.[39]

Across studies, care was actualised via 'self-management', self-investigation and 'hypothesising' about Long COVID illness. For instance, some studies described the management of Long COVID through processes of embodied experimentation, such as trying different diets, pharmacological interventions, vitamins and supplements or alternative therapies.[31 33 35 38 43 44] Many participants described managing their illness as a series of 'trade-offs' or 'compromises' where they worked out what they had the energy to do each day.[30 31 33 35 39 40 44] These experiments required participants to monitor their bodies to determine what was working or what could be triggering particular illness effects. Several participants also engaged in collective 'sense-making' and knowledge generation via online Long COVID communities.[31 32 37 40 41] While these communities provided practical support and validation, participants in one study highlighted the potential dangers in these cultures of knowledge sharing, which could include 'unverified medical advice'.[41]

## Recovery

Studies presented the experience of Long COVID recovery as marked by 'uncertainty'.[31–33 37–39 41 45] While some participants described recovery as an incremental process towards becoming well again,[33 44] far more studies included participants who described recovery in terms of 'episodic illness' or 'disability'.[33 39 40 44] In these cases, the illness became 'unpredictable' in nature and was marked by 'relapses' or periods of worsened symptoms.[32–35 37 39–41 44 45] This episodic experience of Long COVID disrupted narratives of recovery as a 'linear pathway' from 'sick' to 'better'.[37 39–41 43 45] Participants who had experienced Long COVID earlier in the pandemic found this particularly troubling, as their lived experiences of Long COVID differed greatly from the public understanding of COVID-19 illness (where the danger was perceived to lie in whether or not the acute illness caused death).[41 43] Some participants also described an initial period of 'recovery' following acute illness or periods of 'partial recovery' from Long COVID, which troubled this trajectory further.[35 40]

The chronicity of recovery raised ontological questions about the relationship between acute illness and Long COVID. Long COVID was characterised by some participants as existing on a 'continuum' from the time of acute illness, while others saw it as 'fluctuating' and organised as 'clusters' or 'coming in waves'.[35 39 40] One study described conflicting participant perspectives as to whether Long COVID even existed as a discrete illness or condition; some participants instead described their experiences as simply a period of recovery from COVID-19, as one might expect following any flu-like illness.[35] Others described Long COVID as an especially significant experience of recovery that might not ever result in a full return to their previous health.[41] The separation between recovery from acute illness and Long COVID as a distinct condition was complicated further when relapses resulted in hospitalisation, which could produce additional illness events from which to recover.[45]

The terminology of Long COVID also related to how recovery was characterised. For some participants, labels like 'Long COVID' validated their experiences as distinct from other illnesses (such as acute COVID-19 or ME/CFS) and as a recognisable illness shared with other people.[34 35 37 38] However, one study described some participants feeling as though the label implied that their experiences were part of the *same* illness as acute COVID-19.[35] Other languages rendered the chronicity and ontology of Long COVID similarly complex. For example, some participants identified as being 'disabled' or having a 'chronic' illness or health condition.[34 36 39 41 44] These terms designated Long COVID as a stable and ongoing condition rather than something to be recovered from (or a process of recovery in and of itself), which was a shift in framing that some participants found distressing.[35 36]

These experiences all intersected with expectations of what recovery 'should' look like. Participants described social and professional pressures to 'recover' and return to 'normal'.[30 31 34 42] Participants' 'guilt' about being unable to perform their previous social roles was compounded when they additionally required care from their family

or partner.[30 34 42] Some participants internalised these expectations and questioned their own sense of what it means to recover from illness.[37 40 41] One study summed up the conflict between expectations of recovery and the lived experience of Long COVID illness as a tension between the roles of Long COVID patient and 'COVID-19 survivor'.[30]

## DISCUSSION

Our synthesis generates three intersecting analytical themes relating to the epistemic construction of Long COVID: the illness, the patient and recovery. We find that the constitution of Long COVID across these themes produces different implications for the research, management and lived experience of Long COVID. This resonates with other social science literature on the diverse constitution of Long COVID.[4 7 21] One sociological review, for instance, has cautioned that flexible approaches to describing and diagnosing Long COVID can produce challenges for developing pragmatic care information and targeted health interventions.[5]

Our discussion responds to this critique by accentuating the importance of flexible *yet situated* approaches to understanding Long COVID. We call attention to the many ways that Long COVID has been constituted *in* and *by* studies and the effects of this constitutive work. This underscores the importance of reflecting on how the methods and approaches of qualitative studies themselves shape the understandings of Long COVID they produce.[22] We conclude by reflecting on the emergent fluidity and diversity of lived experience in qualitative accounts as defining characteristics of the Long COVID condition. In doing so, we highlight the need for adaptive strategies in diagnosis and care.

### Reflections on study design

The design and methods of qualitative study constitute the object of Long COVID, and thus, how Long COVID comes to be understood, in specific ways. Studies variably framed Long COVID as a diagnostic category, a lived experience of illness or recovery, a social identity, a medical object of research or care, a policy concern and more. Studies also defined Long COVID in relation to health guidelines and/or accounts emerging through patient advocacy, which diversely characterised Long COVID; for example, Box 1 presents descriptors that were *applied to* and *demarcated from* definitions and accounts of Long COVID across the studies. Inconsistencies between these characterisations of Long COVID (as well as across lived experiences and clinical interpretations described within study data) were generally navigated by framing Long COVID as an 'uncertain', 'contested' or 'emerging' illness.[33 35–38 41 43 44]

The constitution of Long COVID within participant eligibility criteria similarly varied across study methods. Most studies recruited participants through Long COVID support groups or by publicly advertising the study, though

---

### Box 1  Descriptors used in Long COVID studies

⇒ Acute COVID-19 illness.
⇒ Recovery following an acute COVID-19 illness.
⇒ Post-COVID syndrome or a similar formal diagnosis.
⇒ Extended effects of acute COVID-19 illness, hospitalisation and/or rehabilitation including being unable to exercise or being on a ventilator.
⇒ COVID-19 symptoms persist for a specified minimum length of time following the onset of COVID-19.
⇒ Symptoms with understood and verifiable underlying biological mechanisms.
⇒ A discrete list of recognised and validated Long COVID symptoms.
⇒ Symptoms of pre-existing illness or disability that are exacerbated following COVID-19 illness.
⇒ Mental illness or psychological distress.
⇒ Chronic illness or disability.
⇒ A syndrome or collective label for many different illness experiences.
⇒ Symptoms that cannot be explained by other diagnoses.
⇒ ME/CFS.

---

some[34–36 45] sampled from clinical settings or existing cohorts, which enabled researchers to identify eligible participants from a broader repository of health data. All recruitment strategies required participants, researchers or clinicians to identify Long COVID patients, which meant that inclusion was shaped by evolving understandings and public awareness of Long COVID at the time of recruitment.

Future qualitative studies need to be more open in their iterative sampling and exploration so as not to close-off or narrow Long COVID as a site that is emergent in its experience. Different qualitative study designs delimit the constitutions of Long COVID differently a priori, which can shape as well as potentially narrow the accounts of lived experience that are produced; furthermore, participant accounts can suggest a greater fluidity and diversity of experience than a priori study definitions of Long COVID imply. This points to the limits of narrow definitions of Long COVID at the outset of studies and the need for study designs to remain attentive to the more open accounts of lived experience that studies may produce.

We suggest that researchers should carefully consider how Long COVID is defined and understood beyond clinical and biomedical understandings. Several studies in our synthesis designed eligibility criteria in terms of self-identification.[33 35–37 40 41 44 45] However, there was limited reflection on what might influence individuals to feel included or excluded from illness categories and how information about illness is accessed and understood by publics. Future study designs that use self-identification might therefore pursue multiple recruitment strategies that use different language and descriptions to invite diverse experiences of the illness, health, or care concern being studied. A defining characteristic of the condition of Long COVID is its inherent fluidity, diversity and multiplicity, and thus, it is important for future studies to avoid

building into their designs a priori definitions of Long COVID that are overly narrow, given how this can shape inclusion, recruitment and accounts of illness experience.

We also emphasise a more general need for greater transparency and reflection on how conceptualisations of Long COVID shape study design, recruitment and analysis. We found, for instance, that even studies with highly specific inclusion criteria extrapolated their findings to more loosely defined and diverse accounts of Long COVID that emerged in their analysis. It is important that future studies make transparent how their conceptualisations of Long COVID shape the constitutions of Long COVID produced. Similarly, studies should ensure epistemological consistency in how they frame and conduct their analyses. A general observation is that qualitative research in this emergent field is relatively under-theorised.

Studies in our synthesis largely employed 'realist' and/or 'critical realist' approaches and conducted descriptive and/or interpretive thematic analyses of semi-structured interview data. A minority of synthesised studies adopted 'critical' and/or 'social constructionist' approaches,[33 36–38 40–42] focusing, for instance, on how narratives 'enact' or 'perform' constitutions of Long COVID.[33 36 37 40 41] There is greater scope for critical sociological qualitative analyses in this field—in particular, analyses that treat narrative as a process of enactment for study, rather than simply a resource of lived experience description.[22 46] The constitution of Long COVID as an emergent, situated and negotiated lived embodiment emphasises the importance of qualitative designs that approach illness, patienthood and recovery as matters of 'construction' and 'enactment'.[22]

### From Long COVID symptoms to Long COVID patients

Across our synthesis, Long COVID was framed as an inherently symptomatic illness experience. Studies we reviewed did not identify *specific* symptoms as proving or precluding a Long COVID diagnosis or experience. Instead, studies accentuated Long COVID as an open category that included symptoms felt to be persistent or recurring following acute COVID-19 infection, symptoms felt to be new and different, as well as symptoms that came and went over time and with varying degrees of severity. Studies used different minimum timeframes of recurring or ongoing symptoms as a condition of Long COVID, yet the 'long' and 'symptomatic' nature of Long COVID was consistently identified as a defining feature of the illness experience and without a clear end point.

Another defining feature across studies was the constitution of Long COVID as a disruption to the 'normal'. Studies noted, and some deployed in their inclusion criteria,[39 42 44] Long COVID symptoms as unexplainable by alternative diagnoses. However, we found that otherwise familiar symptoms, including those from existing medical or health conditions,[31 35 39 45] were made 'strange' in studies through complex symptom clusters, heightened illness effects and unpredictable recovery trajectories that disrupted 'normal' life.[37 40 41 43 44] Furthermore,

the boundaries between the effects of Long COVID as a biomedical illness and the complex social worlds of patients (which included experiencing a global pandemic) were often fuzzy and variably defined.[30–32 34 35 38 39 44 45] This fuzziness made responding to and caring for Long COVID (in formal and informal settings) difficult and could result in participants feeling they had been denied Long COVID patienthood, especially when symptoms were excluded from diagnostic consideration.[31–33 36–41 43]

These inconsistencies and complexities in Long COVID criteria have been subject to critique.[4 5] Yet, our findings suggest that many of the challenges related to more open and changeable constitutions of Long COVID emerge as a result of insufficiently situating this constitutive work. There is a need for researchers and care providers to embrace broader and more open understandings of Long COVID 'symptoms' and 'diagnosis' that reflect the diversity of experiences examined across qualitative studies—not to capture more people within a singular definition of the 'Long COVID patient' with standardised 'care' needs, but to acknowledge the complex reality of Long COVID experiences. For example, our synthesis shows that a Long COVID diagnostic label can provide validation and help individuals make sense of their illness experience even in the absence of clinical care pathways.[31–33 35 37 40 41 43] This emphasises the clinical and social value of Long COVID diagnosis beyond its capacity to support linkage to care or treatment.

### The object of Long COVID recovery

Like Long COVID patienthood, recovery is multiply interpreted across the studies we reviewed. Our synthesis suggests that non-linear trajectories of Long COVID recovery, where illness does not become progressively better over time but is instead marked by unexpected or relapsing symptoms, are experienced as a disruption of recovery itself.[39–41 43 45] This produces new health uncertainties when expectations of what Long COVID recovery 'should' look like fail to materialise.[30 31 34 37 41–43] Across studies, these effects were most noticeable for participants who had experienced Long COVID earlier in the pandemic when there was less knowledge about post-COVID-19 illness and recovery.[32 33 37 40 41 43]

The non-linearity of Long COVID experiences also challenged notions of 'full' recovery or 'partial' recovery. Some participants experienced COVID-19 recovery as a separate process to Long COVID (recovery),[35 40] and others resisted assumptions of recovery entirely, describing themselves as disabled.[34 36 39 41 44] Here, our synthesis resonates with existing research on experiences of chronic illness, which argues that the so-called 'restoration' of bodily capacities following illness and disability is not always realistic or possible.[47 48] Furthermore, while diagnostic labels can give patients 'permission to be ill',[49 50] our findings suggest that this social permission is predicated on the assumption that illness is temporary and that patients will soon resume 'normal' professional and social activities.[30 31 34 42] Such conditional freedom from

social obligations also mirrors broader experiences of chronic illness and disability.[47 51 52]

As observed in other social and health research contexts,[53–55] diverging understandings of what constitutes illness and recovery produce challenges for designing research and health interventions. Our analysis identifies multiple health 'problems' from which participants were described as recovering, including SARS-CoV-2 viral infection, COVID-19 illness (and in some cases hospitalisation), Long COVID or something less precise and entangled with broader social factors and the pandemic context. Processes of recovery were also diversely characterised. While a 'return to normal' was both possible and desirable for some, recovery was also characterised as an unpredictable process requiring ongoing care, a process of working towards a new version of normal, or was rejected entirely.

These insights highlight the importance of interrogating the problem and goal of care and recovery when discussing Long COVID. Notions of care, patienthood, recovery and being 'recovered' held different meanings for participants across studies, which had implications for how the experience of Long COVID became understood—for participants, their clinicians, people in their social and professional lives and through the framing and analyses of the studies themselves. While one person's definition of being recovered from (Long) COVID might require the complete absence of symptoms or illness effects, others might feel recovered when they feel well enough to work or no longer require frequent medical interventions.

Rather than standardising clinical definitions of Long COVID,[56] we propose that practical approaches to the study and care of Long COVID be developed at the intersections of emergent illness, care and recovery experience. Our recent study of COVID care practice suggests that situating uncertain illness in this way is not only possible but is in many ways, a familiar element of adaptive healthcare practice and research that can be built upon.[57]

## Limitations

Qualitative research on Long COVID has emerged alongside rapidly evolving clinical understandings of COVID-19 and post-COVID illness. We have highlighted inconsistencies within and between studies regarding what counts as 'Long COVID'. There is an opportunity for comparative approaches to qualitative research that characterise how Long COVID is differently understood in relation with illness experience, care needs and recovery expectations. We have also noted a need for greater consistency and reflection within studies regarding how they frame, design and analyse the accounts of Long COVID they produce. Many qualitative studies included in our synthesis were developed as adjuncts to surveys, cohorts and/or clinical studies rather than as stand-alone qualitative studies. There is scope for greater theoretical, sociological and critical methodological engagement in this emergent field of qualitative study. Included studies primarily draw from the UK and USA and are limited to studies published in English.

## CONCLUSION

We accentuate Long COVID as a site of multiple qualitative interpretation, giving rise to coexisting versions of illness, patienthood and recovery. Crucially, we find that Long COVID develops meaning through its social relations of varied and diverse lived experience. We caution against attempts to align or stabilise constitutions of Long COVID illness, patienthood and recovery in a singular trajectory or understanding. Accounts of patients' experiences illuminate how illness, patienthood and recovery are sites of continual negotiation and thus, are also defining qualitative features of what Long COVID is. Efforts to diagnose and intervene must be as adaptive as the varieties of Long COVID lived experience.

**Contributors** MH led the conception and design of the review, data collection and analysis. TR and KL contributed to research design and analysis. MH drafted the manuscript, with all authors contributing to write-up and revision. All authors reviewed and gave final approval of the version to be published and agree to be accountable for all aspects of the work. MH is responsible for the overall content as guarantor.

**Funding** This research was funded by the National Institute for Health Research (grant NIHR135315). The views expressed are those of the authors and not necessarily those of the NIHR or the Department of Health and Social Care. We are grateful for additional support from the UNSW SHARP (TR) and Scientia Fellowship (KL) schemes. KL is supported by an Australian Research Council DECRA Fellowship (DE230100642).

**Competing interests** None declared.

**Patient and public involvement** Patients and/or the public were not involved in the design, or conduct, or reporting, or dissemination plans of this research.

**Patient consent for publication** Not applicable.

**Ethics approval** Ethics approval linked to grant NIHR135315 was granted via the London School of Hygiene and Tropical Medicine Observational Research Ethics Committee (Ref 28635).

**Provenance and peer review** Not commissioned; externally peer reviewed.

**Data availability statement** All data relevant to the study are included in the article or uploaded as supplementary information. Not applicable.

**ORCID iD**
Mia Harrison http://orcid.org/0000-0001-8629-9901

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
