## [Reviewer comments · BMJ Open]

ARTICLE DETAILS

TITLE (PROVISIONAL)	The constitution of Long Covid illness, patienthood, and recovery: A critical synthesis of qualitative studies
AUTHORS	Harrison, Mia; Rhodes, Tim; Lancaster, Kari

VERSION 1 – REVIEW

REVIEWER	Humphreys, Helen Sheffield Hallam University, Advanced Wellbeing Research Centre
REVIEW RETURNED	22-Jan-2024

GENERAL COMMENTS	I commend the authors on their attempt to synthesise the increasingly extensive qualitative literature on Long Covid. I suggest that this paper could go further in its recommendations for future research in this field and be more explicit about what these findings add. I have suggested some revisions below which I hope are useful to strengthen the quality and significance of the paper. Methods Can the authors provide more detail on the process of theme/theory development? E.g. how many researchers contributed to theme development, how were these negotiated/shaped over the course of the synthesis, how many stages of refinement did they go through etc. In terms of transparency, reflexivity – what was the background of the authors, how might this have affected theme development and/or how was this critically reflected on? Discussion “Future qualitative studies need to be more open in their iterative sampling and exploration so as not to close-off or narrow Long Covid as a site which is emergent in its experience, including through how it is defined and understood beyond clinical and biomedical understandings.” I understand the reasons behind this recommendation but can the authors explain how this might be practically achieved and/or how traditional research processes might need to be adapted to enable this? For example, most researchers find they need to define the phenomenon they are studying in order to develop inclusion criteria and/or appropriate recruitment strategies approved via ethical review boards – and thus have to use the knowledge available at the time of their study (whether that be clinical/diagnostic guidelines or patient-led definitions). How might a more iterative/exploratory sampling approach work in practice? “There is a need for researchers and care providers to embrace broader and more open understandings of Long Covid ‘symptoms’
--

	and 'diagnosis' that reflect the diversity of experiences examined across qualitative studies—not to capture more people within a singular definition of the 'Long Covid patient' with standardised 'care' needs, but to acknowledge the complex reality of Long Covid experiences." I agree this is justified but can the authors go further in explaining how broadening understandings of Long Covid might benefit those living with it? Also, thinking more critically, what are the potential challenges or downsides of doing so? Within both the background and conclusion, can the authors be more explicit about what this study adds to other qualitative syntheses already published? The suggestion that conceptualisation/description of long Covid both in terms of illness experiences and care responses is varied and complex seems to be well established already. General writing The paper could benefit from a thorough edit to reduce word count and simplify language.
--	--

REVIEWER	Sharma, Gagan Guru Gobind Singh Indraprastha University, University School of Management Studies
REVIEW RETURNED	01-Feb-2024

GENERAL COMMENTS	I am pleased with the idea of consolidating the qualitative interviews of Long Covid cases for illness, recovery and patienthood. My suggestion is to kindly reflect the limitations in the manuscript separately.
--

REVIEWER	Lima, Kassia Universidade do Estado do Amazonas
REVIEW RETURNED	08-Feb-2024

GENERAL COMMENTS	The subject addressed is relevant and of total interest to public health worldwide, with a potential contribution to the area. The manuscript is well written, structured and The methods employed seemed adequate I would highlight the finding regarding the "invisibility" or even "non-recognition" of symptoms by health professionals, according to the reports shared. I recommend further discussion of this relevant finding. Although the abstract and conclusion meet the criteria established by the journal, in my opinion, they did not cover the magnitude of the findings discussed in the manuscript. I recommend improving these topics. The possible limitations of the study are unclear. Finally, I recommend accepting it with a few minor revisions.
--

VERSION 1 – AUTHOR RESPONSE

Reviewer 1 Comments

#	Feedback	Response	Pp.
---	----------	----------	-----

1	I suggest that this paper could go further in its recommendations for future research in this field and be more explicit about what these findings add.	Our revisions in response to Reviewer 1's points 4–6 address this feedback. We have also added a paragraph at the end of Discussion and sentences in the new Limitations section to explicitly recommend directions for future research.	18
2	Can the authors provide more detail on the process of theme/theory development?	Further details have been added to the Data extraction and analysis subsection of the Methodology.	7
3	In terms of transparency, reflexivity – what was the background of the authors, how might this have affected theme development and/or how was this critically reflected on?	We have acknowledged the disciplinary tradition and experience that informed our approach to theory development and analysis in the Data extraction and analysis subsection of the Methodology.	7
4	“Future qualitative studies need to be more open in their iterative sampling and exploration so as not to close-off or narrow Long Covid as a site which is emergent in its experience, including through how it is defined and understood beyond clinical and biomedical understandings.” I understand the reasons behind this recommendation but can the authors explain how this might be practically achieved and/or how traditional research processes might need to be adapted to enable this?	We have revised this section and added specific suggestions for how sampling might be approached in future studies.	15
5	“There is a need for researchers and care providers to embrace broader and more open understandings of Long Covid ‘symptoms’ and ‘diagnosis’ that reflect the diversity of experiences examined across qualitative studies—not to capture more people within a singular definition of the ‘Long Covid patient’ with standardised ‘care’ needs, but to acknowledge the complex reality of Long Covid experiences.” I agree this is justified but can the authors go further in explaining how broadening understandings of Long Covid might benefit those living with it? Also, thinking more critically, what are the potential challenges or downsides of doing so?	We have added a couple of extra sentences here to more clearly point to these benefits. Our revisions in response to Reviewer 1's points 4 and 6 further reflect on the critical considerations for broadening understandings of Long Covid.	16
6	Within both the background and conclusion, can the authors be more explicit about what this study adds to other qualitative syntheses	Our introduction provides an overview of existing reviews and qualitative syntheses of Long Covid, and what	3–4; 13–

	already published? The suggestion that conceptualisation/description of long Covid both in terms of illness experiences and care responses is varied and complex seems to be well established already.	these have contributed to existing understandings of Long Covid. We have revised our introduction and discussion to further emphasise that our synthesis responds to critiques within this existing qualitative literature. These revisions also reinforce our key findings, which are not simply that conceptualisations of Long Covid are varied and complex; rather, we show how different conceptualisations of Long Covid have highly situated utility and implications for care and research, but that these can be lost in qualitative research as a result of restrictive study designs or inconsistent framing.	14; 17– 18
7	The paper could benefit from a thorough edit to reduce word count and simplify language.	We have simplified the language throughout.	N/A

Reviewer 2 Comments

#	Feedback	Response	Pp.
1	My suggestion is to kindly reflect the limitations in the manuscript separately.	We have included a separate Limitations section.	18

Reviewer 3 Comments

#	Feedback	Response	Pp.
1	I would highlight the finding regarding the "invisibility" or even "non-recognition" of symptoms by health professionals, according to the reports shared. I recommend further discussion of this relevant finding.	We have made several revisions throughout our Discussion to highlight how definitions of Long Covid have included and excluded particular symptoms or experiences, as well as the effects of these inclusions and exclusions.	15– 16
2	Although the abstract and conclusion meet the criteria established by the journal, in my opinion, they did not cover the magnitude of the findings discussed in the manuscript. I recommend improving these topics.	The edits and additions to the Discussion section in response to Reviewer 1's points 4–6 further accentuate the implications and contributions of the synthesis.	

		We have also made a minor tweak to the abstract.	2
3	The possible limitations of the study are unclear.	We have added a separate Limitations section.	18

VERSION 2 – REVIEW

REVIEWER	Lima, Kassia Universidade do Estado do Amazonas
REVIEW RETURNED	Universidade do Estado do Amazonas 12-Mar-2024

GENERAL COMMENTS	I ratify the relevance of the subject to global public health. The authors took care to respond to the opportunities for improvement highlighted by the reviewers, with theoretical and conceptual depth and clarification of the study's possible limitations. I noticed the adjustments to the abstract and understood that the study emphasized the theme within the scope of scientific research, provoking important reflections in the area of health care, especially in welcoming and recognizing the health needs of those affected. The authors have improved the writing of the manuscript, so I recommend that it be published.
--